# Fast and Accurate Pneumocystis Pneumonia Diagnosis in Human Samples Using a Label-Free Plasmonic Biosensor

**DOI:** 10.3390/nano10061246

**Published:** 2020-06-26

**Authors:** Olalla Calvo-Lozano, Anna Aviñó, Vicente Friaza, Alfonso Medina-Escuela, César S. Huertas, Enrique J. Calderón, Ramón Eritja, Laura M. Lechuga

**Affiliations:** 1Nanobiosensors and Bioanalytical Applications Group (NanoB2A), Catalan Institute of Nanoscience and Nanotechnology (ICN2), CSIC, CIBER in Bioengineering, Biomaterials and Nanomedicine and BIST, Campus UAB Bellaterra, 08193 Barcelona, Spain; olalla.calvo@icn2.cat (O.C.-L.); cesar.sanchez.huertas@rmit.edu.au (C.S.H.); laura.lechuga@icn2.cat (L.M.L.); 2Institute for Advanced Chemistry of Catalonia (IQAC), CSIC, CIBER in Bioengineering, Biomaterials and Nanomedicine c/Jordi Girona 18–26, 08034 Barcelona, Spain; recgma@cid.csic.es; 3Institute of Biomedicine of Seville (IBiS), University Hospital Virgen del Rocío/CSIC/University of Seville and CIBER in Epidemiology and Public Health, 41013 Seville, Spain; vfriaza-ibis@us.es (V.F.); sandube@cica.es (E.J.C.); 4Institute for Applied Microelectronics (IUMA), University of Las Palmas de Gran Canaria, 35017 Las Palmas, Spain; aescuela@iuma.ulpgc.es; 5Integrated Photonics and Applications Centre, School of Engineering, RMIT University, Melbourne 3001, Australia; 6Department of Medicine, University of Seville, 41013 Seville, Spain

**Keywords:** *Pneumocystis jirovecii*, surface plasmon resonance, optical biosensor, clinical diagnosis, triplex, DNA capture

## Abstract

*Pneumocystis jirovecii* is a fungus responsible for human Pneumocystis pneumonia, one of the most severe infections encountered in immunodepressed individuals. The diagnosis of Pneumocystis pneumonia continues to be challenging due to the absence of specific symptoms in infected patients. Moreover, the standard diagnostic method employed for its diagnosis involves mainly PCR-based techniques, which besides being highly specific and sensitive, require specialized personnel and equipment and are time-consuming. Our aim is to demonstrate an optical biosensor methodology based on surface plasmon resonance to perform such diagnostics in an efficient and decentralized scheme. The biosensor methodology employs poly-purine reverse-Hoogsteen hairpin probes for the detection of the mitochondrial large subunit ribosomal RNA (mtLSU rRNA) gene, related to *P. jirovecii* detection. The biosensor device performs a real-time and label-free identification of the mtLSU rRNA gene with excellent selectivity and reproducibility, achieving limits of detection of around 2.11 nM. A preliminary evaluation of clinical samples showed rapid, label-free and specific identification of *P. jirovecii* in human lung fluids such as bronchoalveolar lavages or nasopharyngeal aspirates. These results offer a door for the future deployment of a sensitive diagnostic tool for fast, direct and selective detection of Pneumocystis pneumonia disease.

## 1. Introduction

*Pneumocystis jirovecii* is an atypical fungus exhibiting pulmonary tropism responsible for human Pneumocystis pneumonia (PcP). PcP is one of the most serious and potentially fatal infections encountered in AIDS patients. However, with the currently rising number of patients receiving immunosuppressive therapies for malignancies such as cancer, allogeneic organ transplantations and autoimmune diseases, PcP is becoming more and more common in non-HIV immunosuppressed individuals [1]. In fact, PcP remains an important cause of morbidity and mortality worldwide with a mortality rate for HIV-infected patients ranging between 11 and 53% [2,3].

The clinical presentation of PcP may differ from HIV-infected patients to other immunocompromised patients, and there are no specific symptoms or signs. Therefore, its diagnosis continues to be challenging [4]. There is no universally agreed approach to the initial management of patients with suspected PcP, and many institutions treat patients empirically, while others pursue a definitive microbiological diagnosis [5].

Traditional diagnosis of PcP relies on clinical and radiological data but confirmation requires microscopic visualization of the microorganism in stained respiratory specimens since they cannot be grown in vitro [6]. Typically, the respiratory specimens are obtained using sputum induction or fiberoptic bronchoscopy with bronchoalveolar lavage (BAL). Sputum induction by inhalation of a hypertonic saline solution is the quickest and least-invasive method for definitively diagnosing PcP. If sputum induction is non-diagnostic or cannot be performed, then a bronchoscopy with BAL is the next step and remains the ‘‘gold standard’’ for diagnosis of PcP [2,4]. Nevertheless, all of the direct organism visualization methods can lead to false-negative results, consequently, a negative sputum induction cannot rule out a diagnosis of PcP [7].

On the other hand, Polymerase Chain Reaction-based (PCR-based) techniques have demonstrated high efficacy to detect *P. jirovecii* DNA from diverse kinds of clinical specimens (BAL, induced sputum, expectorated sputum, oropharyngeal or nasopharyngeal aspirates samples (NPA), biopsy specimens) [7,8]. The sensitivity of the PCR depends on the selected target gene and primers. Nevertheless, a comparison between PCR-based techniques and a staining method such as immunofluorescence proved that PCR-based techniques are more sensitive and close-fitting to the histological evidence [9]. Conventional or real-time PCR assays based on the amplification of the mtLSU rRNA gene from the microorganism are the most commonly used [4,10], but many other sequences have been targeted (major surface glycoprotein, internal transcribed spacers, thymidylate synthase, dihydrofolate reductase, heat-shock protein 70, among others) [2,4,7]. However, comparative evaluating studies are difficult to perform because of different clinical contexts, sampling methods, laboratory reagents or technical strategies used for DNA extraction, amplification or analysis of results [7]. A concrete limit of detection has not been reported for PCR techniques due to the lack of standardization, but some publications consider <10^3^ copies/mL as the limit of detection to diagnostic Pneumocystis pneumonia [9,11]. In addition, these PCR-based techniques are time-consuming and require specialized personnel and instruments. Identification of patients having PcP and classification into mild, moderate or severe disease could provide a guide for the choice of the most suitable drug for treatment, as well as assist in deciding if adjuvant corticosteroids are indicated [12].

Since PcP can be rapidly progressive and the mortality rate remains high, early therapy is essential. Efforts focused on rapid, portable and low-cost techniques are needed for an early diagnosis, prevention and clinical response to PcP [4]. In this sense, label-free optical biosensors, and specifically those based on plasmonic technology, are excellent alternatives as analytical tools with great potential for point-of-care diagnostics [13,14]. Plasmonic biosensors are characterized by high sensitivity, versatility and capability for multiplexed detection and miniaturization. They can also be implemented in a user-friendly scheme, making them very attractive as clinical diagnostic tools. Among plasmonic biosensors, the surface plasmon resonance (SPR) biosensor is the most matured one and has been widely commercialized by diverse companies worldwide, and is employed routinely in research laboratories and in pharmaceutical industries for the study of virtually any type of biomolecular interaction analysis [14,15]. The sensing mechanism is based on the generation of an optical surface plasmon by coupling a light to a thin metal layer (usually 45–50 nm thick gold). This optical plasmon wave generates a strong evanescent field very sensitive to minute refractive index changes at the surface of a gold sensor chip, such as those generated by the ion interaction of the analyte with an immobilized bioreceptor at the sensor surface. These variation in the refractive index produce an angle or wavelength variation of the reflected light, which is directly related to the amount of analyte bounded on the bioreceptor surface. SPR has already proven to be an extremely useful tool for nucleotide analyses [15,16]. In most of the cases, the sensing process is based on the recognition of single DNA strands by forming a duplex structure with their complementary probe strand, which is previously attached to the surface of the sensor chip. DNA-based probes can also interact in different ways with their target molecules. For example, other DNA-based structures such as aptamers adopt a determined complexion that is able to recognize their corresponding protein analytes [17,18]. Recently, triplex forming DNA probes have been gaining increasing attention due to their ability to form nucleic acid triplexes by the association of three nucleic acid strands. They are formed by the addition of a third strand to a duplex, containing tracks of polypurine-polypyrimidine sequences [19]. The design, synthesis and use of hairpins for the formation of triplexes in biosensing and gene inhibition have been recently reviewed [20,21]. A marked enhancement for the detection when using the triplex structure configuration compared with the conventional duplex approach has been reported, for example, for the detection of miRNAs for cancer diagnosis [22] or RNAs from Listeria innocua with predicted secondary structures [23,24]. More recently, polypurine reverse-Hoogsteen hairpins (PPRHs) formed by two antiparallel polypurine mirror strands have been used as bioreceptors for DNA methylation analysis using a SPR biosensor [25,26].

We have implemented here a SPR biosensor for the detection of Pneumocystis pneumonia. For that, we have designed a specific PPRH probe able to detect the mtLSU rRNA gene from *Pneumocystis jirovecii*, with sensitivities of 2.11 nM. We have demonstrated, for the first time, how our innovative PPRH probe can capture double-strand DNA (ds-DNA) from lung fluid samples and can diagnose PcP in a direct, label-free and fast way, without any PCR amplification and using low volumes (50 µL) of human samples (Figure 1). Our biosensor is demonstrated to be an efficient screening tool of PcP in a fast, user-friendly and non-invasive way, evidencing its potential to be employed as the preferred diagnostic solution for PcP diagnosis.

## 2. Materials and Methods

### 2.1. Synthesis of the Oligonucleotides

The PPRHs were designed to carry two antiparallel polypurine sequences (green and blue) complementary to the pyrimidine region of the gene (red) encoding the mitochondrial large subunit ribosomal RNA of *Pneumocystis jirovecii*, to form the antiparallel triplex structure. The purine part of the hairpin is connected head-to-head with the reverse-Hoogsteen sequence using a tetrathymidine sequence. For biosensing purposes, the oligonucleotides were prepared as described with an additional 15 thymines (T_15_) and Thiol-Modifier C6 S-S CE Phosphoramidite (Link Technologies, Bellshill, Scotland) in the 5′-end (Figure 2a). The complementary sequence to form a duplex structure is designed to carry the polypurine sequence (green) with additional bases complementary to the analyte (red) (Figure 2b). Finally, a control hairpin was prepared using the same strategy but the polypurine sequence of the reverse-Hoogsteen has an incorrect sequence (pink) that prevents the formation of a triplex structure (Figure 2c). Sequences were prepared on an Applied Biosystems 3400 (Applied Biosystems, Foster City, CA, USA) synthesizer using controlled-pore supports (scale 1µM) according to the protocols of the manufacturer. Standard protecting groups were used for DNA sequences (A^Bz^, G^ibu^, C^Bz^, T). After assembling of the sequences, oligonucleotide supports were treated with aqueous ammonia (32%) for 16 h at 55 °C with 0.1 M DL-Dithiothreitol (DTT). The resulting solutions were evaporated and used directly.

### 2.2. Preparation of the Samples

BAL or sputum specimens as NPA were obtained from patients with PcP or chronic obstructive pulmonary disease (COPD) admitted to Virgen del Rocio University Hospital (Seville, Spain). DNA from respiratory specimens was extracted and purified using a Nucleospin Tissue Kit (Macherey-Nagel, Bethlehem, PA, USA) after digestion with proteinase K at 56 °C and conserved at −80 °C for further assays.

Detection of *P. jirovecii* DNA was done using nested-PCR amplification of the Pneumocystis mtLSU rRNA gene, as described elsewhere [27]. Identification of other fungi was performed, amplifying the fungal nuclear ribosomal internal transcribed spacer ITS2 region [28] using a semi-nested protocol using the primers ITS-1(5′-TCCGTAGGTGAACCTGCGG-3′) and ITS-4 (5′-TCCTCCGCTTATTGATATGC-3′) in the first PCR round and ITS-3 (5′- GCATCGATGAAGAACGCAGC-3’) and ITS-4 in the second PCR round; subsequently, the amplification product was cloned and sequenced. *Pseudomonas aeruginosa* presence was checked using conventional PCR targeting the oprL gene using the primers PAO1 S (5′-ACCCGAACGCAGGCTATG-3′) and PAO1 A(5′-CAGGTCGGAGCTGTCGTACTC) [29].

We purified the PCR amplification product (PCR clean up and gel extraction kit of Macherey-Nagel (Bethlehem, PA, USA) from the first and second round of mtLSU rRNA nested-PCR for “short-sequence triplex control detection”. In addition, we produced clones of these sequences using the pGEM-T Easy Vector System (Promega, Madison, WI, USA) in JM109 High Efficiency Competent Cells selected on Lysogeny Broth (LB)/ampicillin/Isopropyl β-D-1-thiogalactopyranoside (IPTG)/5-bromo-4-chloro-3-indolyl-β-D-galactopyranoside (X-Gal) plates, cultivated in LB/ampicillin medium and plasmid purified with NucleoSpin Plasmid (Macherey-Nagel, Bethlehem, PA, USA). To linearize vectors, some of the purified plasmids were digested with EcoRI (Promega, Madison, WI, USA) in buffer H for 1 h at 37 °C and subsequently diluted to the desired concentration.

Pipettes with filters were used in all stages. DNA extraction, preparation of the PCR reaction mixture, ligation, cloning and sequencing were performed in different areas and under laminar air flow hoods or PCR workstations. To detect any cross-contamination, all DNA extraction and PCR reactions were performed with a negative control of sterile water.

### 2.3. Experimental Procedure for Detection Using the SPR Biosensor

#### 2.3.1. Chemicals

See Appendix A.

#### 2.3.2. SPR Biosensor Device

We utilized a portable custom-made SPR sensor. The device is based on the Kretschmann configuration and works at a fixed angle of incidence (*θ* = 70°). The SPR sensor monitors the binding events in real time by tracking the SPR-wavelength displacements (Δλ). For more information, see Appendix B and Figure A1.

#### 2.3.3. SPR Sensor Chips Fabrication and Cleaning

See Appendix C.

#### 2.3.4. Bioreceptor Immobilization

Assays were performed to test the different DNA probes (Figure 2). Each DNA probe was modified at the 5′-end of the reverse-Hoogsteen track with a thiol group for their attachment to the gold surface by chemisorption. A T_15_ was included as a vertical spacer to separate the recognition sequence from the sensor surface and to improve the accessibility of the mtLSU rRNA sequence to the bioreceptor monolayer [10]. In addition, we used CH_3_-PEG-SH (2000 MW) as a lateral spacer to minimize steric hindrance and also to increase the mtLSU rRNA accessibility [25] (Appendix D, Figure A2a).

Clean gold sensor chips were placed on the biosensor and biofunctionalized via thiol-gold chemistry. Gold sensor chips were coated in-situ by flowing the immobilization solution through the sensor flow cell (1 μM DNA bioreceptor with 1 μM CH_3_–PEG–SH and 1 μM TCEP diluted in PBS 50 mM and previously incubated at 70 °C, 650 rpm for 20 min). During the immobilization process, thiolated bioreceptors arrange themselves spontaneously into a so-called self-assembled monolayer (SAM).

Target hybridization was performed at room temperature (RT) and monitored in real time. Different concentrations of the mtLSU rRNA target (5–200 nM) were diluted in 2.5 X SSC buffer + 5% fomamide (FA) and injected into the SPR sensor device at a constant flow rate (18 µL·min^−1^). For more information about the assay optimization, see Appendix D, Figure A2.

In order to dissociate the hybrids and regenerate the sensor surface to allow the analysis of a high number of interactions, a solution of NaOH 5 mM was injected for 60 s at the same rate (18 µL·min^−1^).

For the optimization of the evaluation of the real samples, the analysis was performed by flowing different concentrations of mtLSU rRNA analyte (50–1000 nM) in diethyl pyrocarbonate (DEPC)-H_2_O at a constant flow rate (18 µL·min^−1^).

#### 2.3.5. Data Analysis

See Appendix E.

## 3. Results and Discussion

### 3.1. PPRH Probe Design

First, we analyzed the mtLSU rRNA gene with the aim of searching for homopurine-homopyrimidine tracks susceptible to forming stable triplexes using the Triplex-Forming Oligonucleotide Target Sequence Search Tool of the University of Texas MID Anderson Cancer Center (Austin, TX, USA) [30]. We found contiguous homopurine-homopyrimidine track sequences with the restriction of at least ten nucleotides. Two sequences were obtained. The first one was a purine track target of 13 nucleotides with a single mismatch. The second sequence contained 17 nucleotides, the maximum for a purine consecutive sequence, with three mismatches. We decided to choose the first sequence in order to keep the number of mismatches to a minimum.

PPRHs are able to recognize in a sequence-specific manner polypyrimidine analyte sequences in ds-DNA via Watson-Crick bonds, by producing a triplex structure and strand displacement of the ds-DNA.

We have designed a PPRH probe consisting of two antiparallel poly-purine domains connected by a tetra-thymidine loop. The PPRH probe was designed to capture the complementary homopyrimidine analyte sequence of the fragment of interest. The PPRH is formed by a homopurine strand that hybridizes the analyte sequence using Watson-Crick hydrogen bonds (WC track) and the inverted homopurine portion of the oligonucleotide that forms a triplex helix, by reverse-Hoogsteen hydrogen interactions (RH track). The capture of the analyte pyrimidine sequence using the PPRH produces a strand displacement of the gene mtLSU rRNA in the region of interest (Figure 3).

Antiparallel triplex formation with the PPRH and the target pyrimidine sequence of the mtLSU rRNA gene was studied using Circular Dichroism spectrometry (Appendix F, Figure A3) and compared with duplex. CD of triplex show positive bands at 275 nm with a shoulder at 271 nm and negative bands at 248 and 210 nm.

The synthesis of the PPRH for biosensing purposes is completed after the reverse-Hoogsteen homopurine moiety using a sequence consisting of poly-thymidines (T_15_) that function as a vertical spacer between the PPRH and the end thiol functional group for gold sensor surface coupling.

Two control probes were also designed: a duplex containing a complementary sequence to the analyte pyrimidine in order to form a duplex and a PPRH control that consists of the complementary sequence and a random sequence instead of the correct reverse-Hoogsteen strand that prevents the formation of the triplex. As previously, for biosensor purposes, both oligonucleotides were modified with a T_15_ vertical spacer and a thiol group.

### 3.2. Biosensor for the Diagnosis of PcP

We employed the SPR biosensor and the methodology shown in Figure A1 and Figure 1, respectively, for the diagnosis of PcP. Our specific PPRH probes were attached to the gold sensor surface and were capable of interacting with the ds-DNA contained in the BAL and NPA lung fluid samples. PPRH probes identified specifically the mtLSU rRNA gene by forming a triplex structure due to the strand displacement. The SPR biosensor is very sensitive to refractive index (RI) changes taking place within the evanescent field (Figure A1b), detecting biomolecular interactions in real time and quantifying the concentration of DNA in the sample, by observing the wavelength displacement (Figure A1c). Therefore, in positive samples, the mtLSU rRNA sequence from *P. jirovecii* will interact with our probe, producing an increment in the RI and shifting the resonance curve to higher wavelengths. The tracking of the resonance peak (Δ*λ*) can be followed, making it possible to detect interactions in real time.

As a first step, we evaluated the selectivity and efficiency of the PPRH probe for the detection of the mtLSU rRNA sequence by employing a single strand DNA (ss-DNA) of this gene as analyte. The performance of the PPRH probe was compared with the control duplex formation capture (complementary probe). In addition, we studied the PPRH control probe in which the triplex could not be formed properly. To assess the performance of each bioreceptor, they were immobilized in-situ using a 1:1 ratio (PPRH probe: CH_3_–PEG–SH) on the sensor surface and we monitored their response to the flow of samples containing different ss-DNA mtLSU rRNA concentrations, ranging from 5 to 200 nM in triplicates (Figure 4a). As can be observed in the figure, increasing concentrations generated a sensor signal with increasing resonance shifts, which demonstrates that the sensor responded to the presence of mtLSU rRNA in a concentration-dependent way and it can be used for quantitative analyses. The specificity of the assay was confirmed by evaluating a different sequence with identical length and CG content (Table 1). The presence of 5% formamide in the hybridization buffer eliminated non-specific interactions between the PPRH probe and the DNA control sequence, although it is identical in length and GC content to the mtLSU rRNA sequence (Figure 4b).

The limit of detection (LOD) for each probe was calculated as the concentration corresponding to the blank signal plus three times its standard deviation. The LOD of the PPRH was 2.11 nM (*R^2^* = 0.9706). LODs calculated from the complementary and the control PPRH were 3.14 nM (*R^2^* = 0.9576) and 4.40 nM (*R^2^* = 0.9246), respectively.

According to these results, the triplex helix capture approach enabled a more sensitive detection, recognizing lower concentrations of ss-mtLSU rRNA analyte by the formation of a triplex helix. The complementary probe also detected ss-mtLSU rRNA analyte fairly, but the duplex hybridization achieved through Watson-Crick bonds was not as strong as the link generated by the triplex helix. Finally, the control PPRH probe showed a poorer performance, achieving the worst limit of detection. We suggest that the fact of having non-symmetrical two-polypurines sequences (even if one is complementary and antiparallel to the mtLSU rRNA analyte) avoided the formation of a triplex helix but might also interfere in the complementary hybridization for the duplex approach with Watson-Crick bonds.

To corroborate these results, the calibration curves, shown in Figure 4a, and their equations, described in the data analysis section in Appendix E, have been studied. They not only enable the sensitivity of the interactions to be evaluated through the limit of detection but also the saturation of the number of bioreceptors in the sensor surface (*Bmax*) and their affinity with the analyte (*Kd*).

*Kd* and *Bmax* (Table 2) provide information about the affinity and the number of bioreceptors on the sensor surface, respectively. The lower equilibrium binding constant implies a faster recognition event and, hence, a greater affinity between receptor and analyte. Comparing the *Kd* of each bioreceptor, there were not substantial differences between the PPRH (*Kd* = 44.06 nM) and the complementary probe (*Kd* = 43.36 nM), therefore, the PPRH probe had a similar capture capability to the duplex for ss-DNA mtLSU rRNA detection. However, the complementary probe provided a very slightly better affinity since the recognition only involved the creation of duplex hybrids and not the triplex helix, which is a more complex structure. In contrast, the control PPRH (*Kd* = 78.76 nM) achieved the worst affinity, since it did not generate the triplex helix, although the complementary part was able to detect the mtLSU rRNA analyte. As previously suggested, the non-symmetrical polypurine sequence in the control PPRH could interfere in the recognition between the complementary sequence and the mtLSU rRNA analyte, deteriorating the interaction. Regarding *Bmax*, it has a similar value in all the cases. This means that the biofunctionalization methodology is reproducible, allowing the attachment of the same number of bioreceptors on the sensor surface independently of the DNA sequence or configuration.

In addition to an adequate sensitivity, two of the most important requirements of a biosensor device are the reproducibility and the accuracy. Both parameters were assessed through the coefficient of variation (CV) intra- and inter-sensor chips (evaluations done using the same SPR functionalized surface and for different sensor surfaces, respectively) (see Appendix G
Table A1). The coefficients of variation were obtained as the ratio of the standard deviation of the mean, expressed in percentages (% CV). CV values for the PPRH and complementary probe were close to the maximum variability recommended for clinical analysis (15%) [31], which reflected a good reproducibility and the suitability of these bioreceptors for mtLSU rRNA detection. Nevertheless, the inter-sensor chip CV was substantially higher in the control PPRH case, which reflected the poor accuracy and reproducibility of the control PPRH for mtLSU rRNA recognition. We conclude that PPRH is the most suitable and appropriated bioreceptor for mtLSU rRNA detection. In addition, this probe can generate DNA strand displacement, directly capturing ds-DNA through the formation of the triplex helix, as demonstrated by Huertas et al. [25].

All the results obtained from the calibration curves are shown in Figure 4. The evaluation of the kinetics parameters (Table 2) and the reproducibility of the assay (Table A1) demonstrated that PPRHs were the ideal bioreceptors to detect the mtLSU rRNA gene. The capability of PPRHs to create the triplex helix allowed them to exhibit better sensitivity and similar affinity to the mtLSU rRNA gene than the complementary probe. Moreover, the property of forming triplex helix was an advantage compared to the traditional duplex approach since PPRHs were able to capture ds-DNA.

### 3.3. Analysis of mtLSU rRNA in Patients’ Samples

In order to demonstrate the capabilities of the SPR biosensor for clinical analysis, the next step was the assessment of human samples. As previously described, *Pneumocystis jirovecii* is involved in pneumonia, a lung disease, and its presence has been demonstrated in pulmonary fluids [2,4]. Hence, we employed clinical samples from bronchoalveolar lavage or nasopharyngeal aspirates. DNA from respiratory samples was extracted, purified and diluted in highly pure water (mili-Q water) in a very low volume (30–50 µL).

Prior to the assessment of the respiratory samples, we analyzed the effect of water on the sensor response and the hybridization event. Water could affect the performance of the PPRH probe for ss-DNA mtLSU rRNA sequence identification. Thus, we monitored PPRH response to samples containing different ss-DNA mtLSU rRNA concentrations, ranging from 50 to 1000 nM in triplicates (Figure 5).

As can be observed in Figure 5, the water flowing on the sensor surface generated a decrease in the real-time sensor signal since water has a value of refractive index less than that of the running buffer. However, the signal returned, finally, to the baseline, proving that there is no interaction with the sensor surface or the monolayer, similar to the case of the DNA control (Figure 5a). Regarding the hybridization event, the absence of salts in these samples could deteriorate the hybridization event. In Figure 5, we observed that the sensor response increased as the mtLSU rRNA concentration increased. Therefore, we were able to identify the mtLSU rRNA analyte specifically without non-specific interactions. Nevertheless, the hybridization efficiency was hindered due to the absence of salts and FA. The sensitivity of the SPR biosensor obtained for mtLSU rRNA detection was reduced by five times, from 2.11 nM to 10.14 nM (*R^2^* = 0.9413). Although the sensitivity had been affected, the reproducibility continued to be extremely adequate, as shown in Table A2, where CV values for inter- and intra-sensor chips are below or near to 15%.

In spite of the decrease in sensitivity due to the sample dissolution in water, we were able to detect the ss-DNA mtLSU rRNA sequence. Previously to clinical sample evaluation, we also analyzed the efficiency of PPRH probes for mtLSU rRNA detection in double strand complexions and more complex DNA structures such as plasmids or ds-DNA fragments. We employed pGEM-T Easy plasmids and other DNA fragments that contained the mtLSU rRNA sequence (Figure 6).

The assessment of plasmids and ds-DNA fragments was carried out by flowing all the sample volume (50 µL) over the sensor surface at a rate of 18 µL·min^−1^. As shown in Figure 6, the sensor signals obtained for plasmid samples are low since the PPRH bioreceptor could not identify the mtLSU rRNA sequence in pGEM-T Easy plasmids due to their large size and molecular weight (3400 bp) and their circular conformation, which means complex structures of DNA. These features hindered the accessibility to the mtLSU rRNA sequence and the strand displacement using PPRH probes. Nevertheless, PPRH probes were able to detect shorter and linear fragments of ds-DNA, which show higher sensor responses in Figure 6a. To reaffirm this fact, we digested some plasmids in order to obtain simpler and shorter DNA structures, and as can be appreciated in Figure 6b, PPRH identified mtLSU rRNA in digested samples, but it could not hybridize with the gene sequence in whole plasmid structures. Therefore, the mtLSU rRNA sequence was more accessible due to the simpler conformation of linear DNA. PPRH probes were able to displace DNA strands and form triplex helix in order to detect ds-DNA containing the mtLSU rRNA sequence.

As we previously suggested, SPR sensitivity was affected and reduced due to the dissolution of samples in water. This is also reflected in Table A3, where the SPR biosensor underestimated the DNA sample concentrations compared to the PCR technique. However, we must stress that the SPR biosensor was able to detect the presence or absence of mtLSU rRNA in ds-DNA samples, and therefore, this biosensor could become a very useful tool for clinical diagnosis.

Different clinical patients’ samples of BAL and NPA were evaluated (Figure 7). These clinical samples included four positive samples for *P. jirovecii*. In addition, eight control samples positive for two other different microorganisms (i.e., *Pseudomones* and *Cladosporium*) were included. To perform the assay, DNA was extracted from BAL or NPA samples, purified, dissolved in highly pure water (milli-Q water), flowed over the biosensor surface at a rate of 18 µL·min^−1^ and measured in real time.

Figure 7 compares the results obtained for each infection based on the determined statistical median of the sensor response. The results of the SPR biosensor assessment of the clinical samples indicated that mtLSU rRNA levels were higher in positive samples compared to negative ones, showing a significant statistical difference in the expression of mtLSU rRNA between patients infected with *P. jirovecii* and the ones infected with others microorganisms such as *Cladosporium* and *Pseudomones* (Figure 7). The positive signal in the SPR biosensor was obtained due to the specific hybridization of ds-DNA mtLSU rRNA to the PPRH probe at the SPR biosensor surface. An ANOVA test with a *p*-value < 0.05 confirmed that mtLSU rRNA from *P. jirovecii* was detected using the PPRH bioreceptor specifically, without cross-hybridization with other microorganism sequences.

The most used methods for the detection of *P. jirovecii* in clinical laboratories include staining and microscopic detection [6] and PCR-based techniques [7,8,9,10,11]. In this paper we demonstrated that a direct detection of mtLSU rRNA gene is possible, without any modification, amplification or labelling, which is related to pneumonia offset. In addition, the methodology was extremely specific and avoided cross-reactivity with other microorganisms. Nevertheless, in order to obtain an adequate and precise quantification of mtLSU rRNA presence in clinical samples, a pre-treatment of the sample should be performed in order to cleavage the DNA and create shorter fragments, improving mtLSU rRNA sequence accessibility and increasing the biosensor sensitivity.

## 4. Conclusions

We have demonstrated the efficiency of a SPR biosensor for the direct and rapid detection of Pneumocystis pneumonia in human fluid samples without amplification or labelling steps in a reduced volume (50 µL). Concretely, we employed PPRH probes, which perform a triplex helix approach to detect ds-DNA, specifically the mtLSU rRNA gene of *P. jirovecii*. The triplex approach ensured better and stronger capture of the mtLSU rRNA gene as compared to the traditional approach based on duplex hybrids through linear DNA probes. The described biosensor methodology allowed mtLSU rRNA to be detected with a LOD of 2.11 nM and was able to discriminate clinical samples for patients infected with *P. jirovecii* from samples infected by other microorganisms such as *Pseudomones* or *Cladosporium*. Nevertheless, more extended studies should be performed to transfer this methodology to a clinical application. The establishment of cleavage protocols as a pre-treatment step of the samples is required in order to obtain shorter DNA fragments. The reduction of the DNA length decreases the presence of complex structures and creates linear fragments, facilitating the accessibility of mtLSU rRNA to the PPRH probes. Thus, the sensitivity of the SPR biosensor for the detection of *P. jirovecii* would improve and similar DNA quantification values would be obtained compared to conventional techniques such as PCR-based ones.

Our SPR biosensor was demonstrated to be an effective and potential tool for PcP clinical diagnosis in a rapid, label-free and user-friendly way. By reducing diagnosis time, it would allow an early clinical response, stopping illness progression and decreasing the mortality rate. It could be being used as the dearest diagnostic tool for PcP diagnosis.

## Figures and Tables

**Figure 1 nanomaterials-10-01246-f001:**
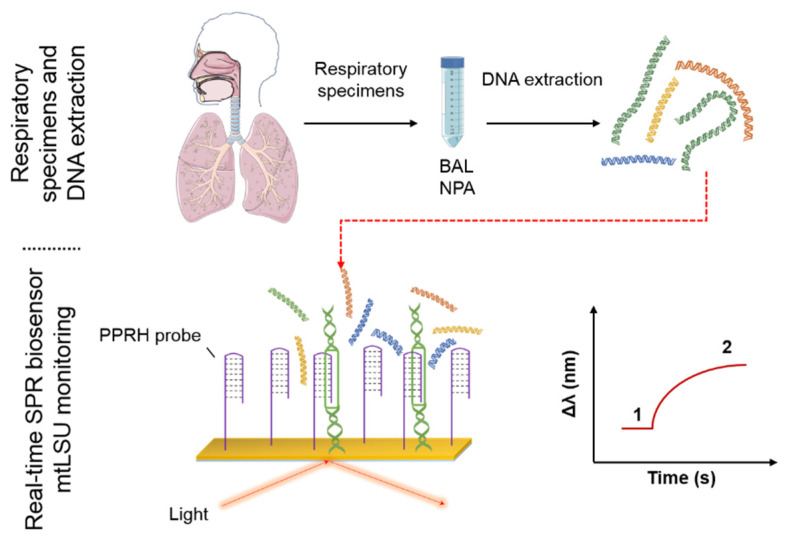
Surface plasmon resonance (SPR) biosensor methodology for the specific detection of *P. jirovecii*. The samples (BAL and NPA) after DNA extraction were injected to the SPR device in which the PPRH probe is attached above the gold surface of the sensor. Wavelength variation of the reflected light is directly related to the amount of the analyte bounded to the bioreceptor. (BAL: bronchoalveolar lavage, NPA: nasopharyngeal aspirates, PPRH: polypurine reverse-Hoogsteen hairpin, 1: Wavelength correspond to the baseline, 2: Wavelength correspond to target analyte recognition).

**Figure 2 nanomaterials-10-01246-f002:**
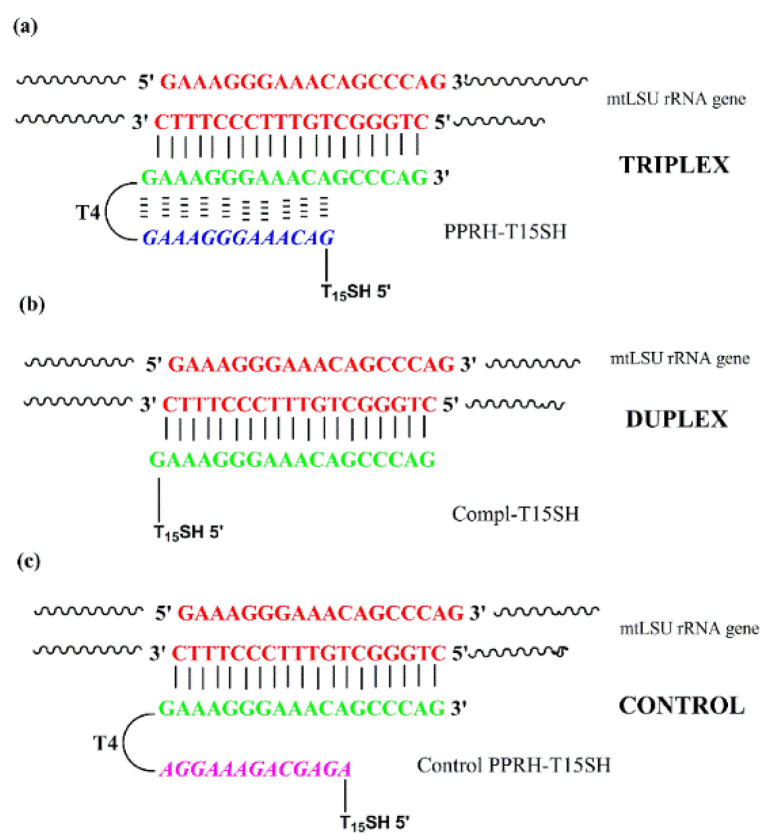
Sequences designed for use as bioreceptors in the SPR biosensor for the detection of the mtLSU rRNA gene of *Pneumocystis jirovecii*. (**a**) PPRH-T_15_SH capture probe recognizes the pyrimidine sequence of the mtLSU rRNA gene and forms a stable antiparallel triplex structure. (**b**) Complementary–T_15_SH forms a duplex structure with the pyrimidine sequence of the mtLSU rRNA. (**c**) Control PPHR-T_15_SH forms a duplex with the pyrimidine sequence of the mtLSU rRNA gene.

**Figure 3 nanomaterials-10-01246-f003:**
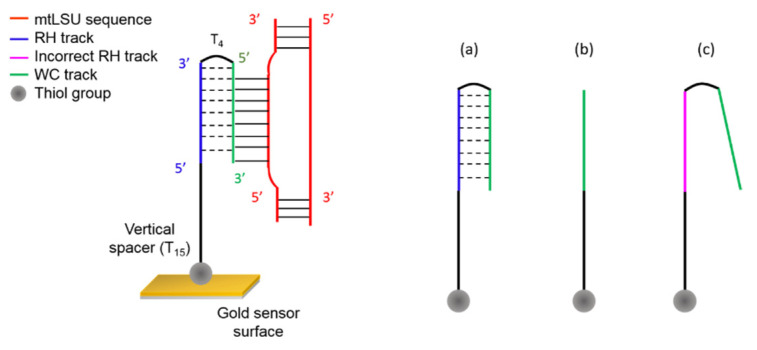
Schematic representation of the mtLSU rRNA gene capture using the PPRH probe, forming an antiparallel triplex structure. In addition, different bioreceptors employed in the SPR biosensor are shown: (**a**) PPRH probe, (**b**) complementary probe, (**c**) control PPRH probe. (Colors correspond to Figure 2).

**Figure 4 nanomaterials-10-01246-f004:**
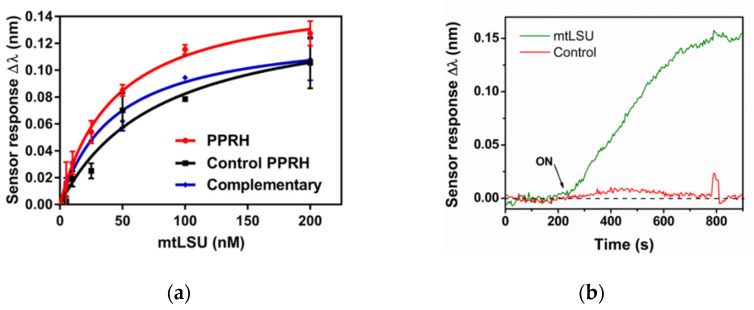
mtLSU rRNA gene detection. (**a**) Calibration curves of mtLSU rRNA on SSC 2.5 X + 5% FA buffer. Sensor response represents the mean ± SD of three measurements in a PPRH: CH_3_–PEG (1:1) receptor monolayer. (**b**) Real-time monitoring of the wavelength displacements (Δ*λ* vs. time) corresponding to the hybridization of 100 nM mtLSU rRNA analyte and DNA control sequences using SSC 2.5 X + 5% FA buffer.

**Figure 5 nanomaterials-10-01246-f005:**
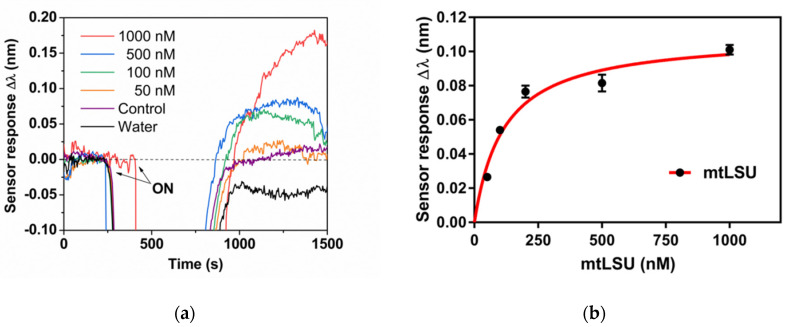
mtLSU rRNA detection. (**a**) Real-time monitoring of wavelength displacements (Δ*λ* vs time) corresponding to the hybridization of different mtLSU rRNA concentrations (1000, 500, 100 and 50 nM, respectively) diluted in water, control DNA and water in a PPRH: CH_3_–PEG–SH (1:1) monolayer. (**b**) Calibration curves of mtLSU rRNA on water using SSC 2.5 X + 5% FA as running buffer. Sensor response represents the mean ± SD of three measurements.

**Figure 6 nanomaterials-10-01246-f006:**
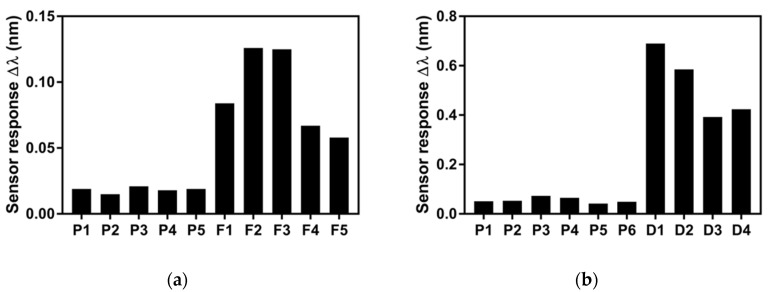
Detection of mtLSU rRNA gene contained in synthetic pGEM-T Easy plasmids and DNA fragments. (**a**) Sensor signal corresponding to pGEM-T Easy plasmids (P) and ds-DNA fragments (F). (**b**) Sensor signal corresponding to pGEM-T Easy plasmids (P) and pGEM-T Easy plasmids digested (D) by EcoRI enzyme. For all the measurements SSC 2.5 X + 5% FA as running buffer and PPRH: CH_3_–PEG (1:1) monolayers were performed.

**Figure 7 nanomaterials-10-01246-f007:**
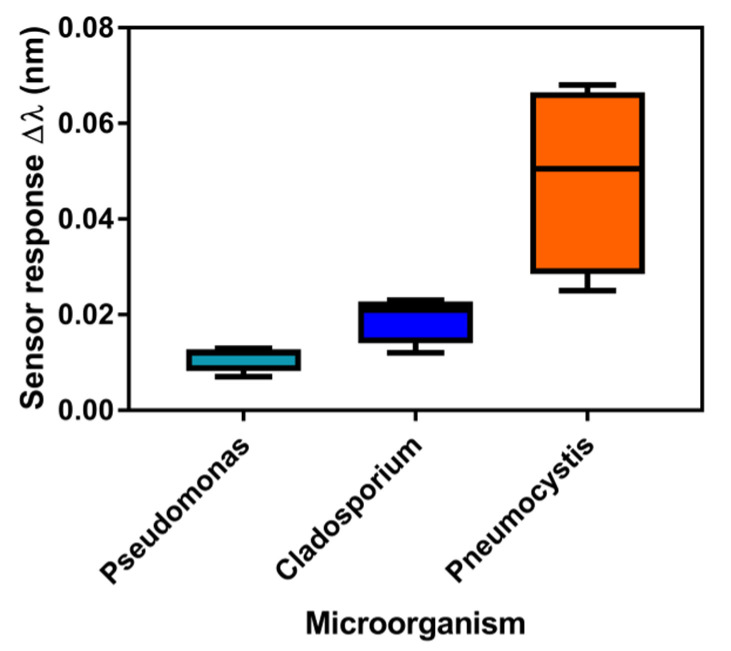
Analysis of mtLSU rRNA in clinical samples from patients infected by (i) *Pseudomones* (*n* = 4), (ii) *Cladosporium* (*n* = 4) and (iii) *Pneumocystis* (*n* = 4), performed using the SPR biosensor biofunctionalized with PPRH: CH_3_–PEG (1:1) monolayer. Representation of one-way ANOVA test where median, maximum and minimum values are shown. One-way ANOVA test, *p*-value = 0.0258.

**Table 1 nanomaterials-10-01246-t001:** Nucleotide sequences, length and CG percentage of the employed mtLSU rRNA analyte and control sequence.

Target	Sequence	Length	GC Content (%)
mtLSU	5′-CTGGGCTGTTTCCCTTTC-3′	18	55.55
Control	5′-TTCCGTGGCTGTTCTCCT-3′	18	55.55

**Table 2 nanomaterials-10-01246-t002:** Kinetics parameters, equilibrium binding constant (*Kd*) and extrapolated maximum number of bioreceptors in the surface (*Bmax*), corresponding to the hybridization interaction mtLSU rRNA analyte—bioreceptor.

	PPRH Probe	Control PPRH Probe	Complementary Probe
*Kd* (nm)	44.06	78.76	43.36
*Bmax* (nm/nM)	0.1593	0.1473	0.1306

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
