# Peer review of "Fast and Accurate Pneumocystis Pneumonia Diagnosis in Human Samples Using a Label-Free Plasmonic Biosensor"

_nanomaterials, 2020, doi:10.3390/nano10061246_

Round 1

Reviewer 1 Report

In their paper Calvo-Lozano et al. demonstrate the use of SPR to diagnose PcP. The system is based on DNA hairpin that interacts with mtLSU rRNA analyte to form a triplex. This consequently leads to change in the SPR signal. The authors demonstrate the feasibility of their system with ss-DNA and with clinical samples. Furthermore, they test their assay with samples from different clinical patients and show the specificity of their system.

The paper is well-written and I really enjoyed reading it! The authors provide enough background that will help the reader to understand the significance of this study, as well as technical details that are missing in many studies. 

My comments are as following:

  1. Additional experiments that show the formation of triplex are required. FRET can be a good way to show it.
  2. The authors should add a comparison in the sensitivity between their method and other methods.
  3. 30% of the references are self-citations! Additional references are needed to provide a broader perspective on the field. Also, whenever the authors refer to previous results, they should provide a reference (For example P9,L:327)
  4. Following the use of mtLSU, the sensitivity of the SPR sensor has been reduced 5 times and not 10 times as the authors wrote (2 nM to 10 nM).
  5. P9: The authors write that they used “highly pure water”. They should explain what does it mean (miliQ water? DDW?).

Author Response

Reviewer(s)' Comments to Author:

Reviewer #1

In their paper Calvo-Lozano et al. demonstrate the use of SPR to diagnose PcP. The system is based on DNA hairpin that interacts with mtLSU rRNA analyte to form a triplex. This consequently leads to change in the SPR signal. The authors demonstrate the feasibility of their system with ss-DNA and with clinical samples. Furthermore, they test their assay with samples from different clinical patients and show the specificity of their system.

The paper is well-written and I really enjoyed reading it! The authors provide enough background that will help the reader to understand the significance of this study, as well as technical details that are missing in many studies.

We greatly appreciate the positive summary evaluation of our work by the reviewer and the suggestions for improvement.

  1. Additional experiments that show the formation of triplex are required. FRET can be a good way to show it.

Thank you for this interesting suggestion. We would like to highlight that the detection of triplex formation using extended antiparallel (Reverse Hoogsteen) purine hairpins is not a simple process for the following reasons: 1) duplex binding is larger than triplex (18 bases implied in duplex formation versus 13 bases in triplex); 2) the dissociation of the triplex strand is not visible by UV (due to the lack of hyperchromicity) and is not pH dependent. We have been working in this subject for the last 18 years, and we have tried several techniques to demonstrate triplex formation with these PPRH, such as gel electrophoresis, CD, aggregation of gold-NP, NMR. Some of these technologies were helpful, but the best and simple method to demonstrate triplex formation is the use of a control sequence with a wrong Hoogsteen strand as we did in the present manuscript. We also run some thermal UV experiments, but we did not observed clear differences as the hypochromicity of the transitions of triplex and duplex for these sequences were similar. However, following the reviewer’s advice, we have included a comparison of the CD spectra of the duplex and the triplex of PcP sequences, and a corresponding comment in the manuscript about such experiment. CD of antiparallel triplex showed positive bands at 275 nm with a shoulder at 271 nm and negative bands at 248 and 210 nm.

We have searched in the bibliography if FRET technology has been employed for the analysis of triplex formation of extended antiparallel (Reverse Hoogsteen) purine hairpins, but we could not find any previous work. Most of the triplex detection by FRET uses parallel triplex that are easier to detect by regular UV-analysis of denaturing curves as they are hyperchromic at 260 nm and hypochromic at 295 nm (due to cytosine protonation). For the present work, we are confident that we have included enough experimental results that demonstrate the triplex formation.  Performing FRET analyses will require a thorough and extended optimization. However, we consider of interest the reviewer’s suggestion and we are really keen to optimize and incorporate FRET analysis for our research in antiparallel hairpins in future projects. 

Paragraph (Page 7, Line 236-238) “Antiparallel Triplex formation with PPRH and the target pyrimidine sequence of mtLSU rRNA gene was studied by CD spectrometry (Figure A3) and compared with duplex. CD of triplex show positive bands at 275 nm with a shoulder at 271 nm and negative bands at 248 and 210 nm.”

 Figure A3 (Page 15, Line 536-540) Appendix A 6.CD spectrometry.

Antiparallel Triplex formation was assessed by circular dichroism in 10 mM sodium cacodylate, 50 mM MgCl2 pH 7.0 buffer.

Figure 3. CD Spectra of Duplex (Compl + ss-DNA mtLSU target) and Antiparallel Triplex (PPRH+ ss-DNA mtLSU target).

  1. The authors should add a comparison in the sensitivity between their method and other methods.

We appreciate the suggestion of the reviewer. In the introduction we explained that the main techniques to detect Pneumocystis pneumonia are microscopic techniques and PCR-based techniques. Although first ones are not DNA quantitative techniques, we have added a section comparing microscopy techniques sensitivities with PCR ones. Regarding to PCR, its sensitivity depends on the target gene as the mtLSU gene, a multicopy gene, or by using a nested PCR. Moreover, there is a lack of reproducibility that affects also the quantitative sensitivity. Therefore, we added a section describing the problems and reported sensitivities of PCR techniques.

(Page2, Line 69-72) “The sensitivity of the PCR depends on the selected target gene and primers. Nevertheless, a comparison between PCR-based techniques and a staining method such as immunofluorescence proved that PCR-based techniques are more sensitive and close-fitting to the histological evidences [9].”

 (Page 2, Line 78-81) “It has not been reported a concrete limit of detection for PCR techniques due to the lack of standardization, but some publications consider <103 copies/mL as the limit of detection to diagnostic Pneumocystis pneumonia [9], [11]”

  1. 30% of the references are self-citations! Additional references are needed to provide a broader perspective on the field. Also, whenever the authors refer to previous results, they should provide a reference (For example P9,L:327)

The referee is right in this observation. We have incorporated 10 new publications of which none are self-citations in order to reduce considerably the percentage of self-citations. All the new citations are included in the References section: Pages 16-18, Lines 551-625.

In addition, we have changed the citations in the text and incorporated those referred to previous results, as referee has suggested. (Lines 60, 66, 69, 72, 74, 76, 78, 81, 85, 90, 96, 103, 108, 111, 112, 115, 116, 118, 159, 160, 166, 195, 328, 334, 344, 424)

  1. Following the use of mtLSU, the sensitivity of the SPR sensor has been reduced 5 times and not 10 times as the authors wrote (2 nM to 10 nM).

We thank the reviewer for pointing out our mistake. We have correct this by replacing “five times” instead “ten times” in the description (Page 11, Line 369)

  1. P9: The authors write that they used “highly pure water”. They should explain what does it mean (miliQ water? DDW?).

According with the suggestion we have added “milli-Q water” (Page 11, Line 406).

Reviewer 2 Report

The submitted manuscript describes SPR-based detection Pneumocystis as a potential replacement for PCR-based methods for clinical diagnosis. Overall I found this to be a well-written and high quality manuscript. The experiments are appropriate and demonstrate a potential clinical use for a well-established technology. I have no major concerns with the methods or presentation of results. Two minor concerns which could be addressed:

(1) The premise that SPR is more user-friendly than PCR is suspect. While PCR requires amplification of DNA, this is routine, and clinical laboratories are already well-equipped for this and contain the necessary instrumentation.

(2) The discussion is abbreviated and intermixed with the results. While this format is fine, the discussion portion of this could be extended somewhat to give better context to these results as they pertain to the larger field.

Author Response

Reviewer #2

The submitted manuscript describes SPR-based detection Pneumocystis as a potential replacement for PCR-based methods for clinical diagnosis. Overall I found this to be a well-written and high quality manuscript. The experiments are appropriate and demonstrate a potential clinical use for a well-established technology. I have no major concerns with the methods or presentation of results.

We appreciate the overall positive evaluation of our work and the constructive feedback from the referee.

  1. The premise that SPR is more user-friendly than PCR is suspect. While PCR requires amplification of DNA, this is routine, and clinical laboratories are already well-equipped for this and contain the necessary instrumentation.

We thank the reviewer for their comment. The reviewer is right affirming that PCR is widely used and performed routinely in the clinic, but it needs to be carefully optimized to maintain similar melting temperatures for the used primers and minimize cross-hybridization potential. In addition, it requires skilled personnel to run the experiments and employs expensive reagents, such as polymerases, which complicates its general use in all the laboratories, especially in developing countries with limited resources [1]. On the other hand, SPR offers a simple and direct detection that usually does not require special skills or sample treatment. This allows the evaluation of real samples (serum, urine…) in an easier way, where interaction between DNA probes immobilized in the surface and DNA target, produces sensor responses in real-time,  reducing the complexity and time of diagnosis.

[1] Oladele, R. O.; Otu, A. A., Richardson; M. D.; Denning, D. W. Diagnosis and management of Pneumocystis pneumonia in resource-poor settings. Journal of health care for the poor and underserved. 2018, 29, 107-158.

  1. The discussion is abbreviated and intermixed with the results. While this format is fine, the discussion portion of this could be extended somewhat to give better context to these results as they pertain to the larger field.

We thanks the reviewer for the advice. According to reviewer suggestion, we have added/modified some paragraphs at the end of sections 3.2 and 3.3 in order to clarify the discussion.

(Page 10, line 335-340) “All the results obtained from the calibration curves showed in Figure 4, the evaluation of the kinetics parameters (Table 2) and the reproducibility of the assay (Table A1) demonstrated that PPRHs were the ideal bioreceptors to detect mtLSU rRNA gene. The capability of PPRHs to create the triplex helix allowed them to exhibit better sensitivity and similar affinity to mtLSU rRNA gene than complementary probe. Moreover, the property of forming triplex helix was an advantage compared to traditional duplex approach since PPRHs were able to capture ds-DNA.”

(Page 12, line 423-430) “The most used methods for the detection of P. jirovecii in clinical laboratories include staining and microscopic detection [6] and PCR-based techniques [7-11]. In this paper we demonstrated that is possible a direct detection of mtLSU rRNA, without any modification, amplification, or labelling, that was related to pneumonia offset. In addition, the methodology was extremely specific and avoided cross-reactivity with other microorganisms. Nevertheless, in order to obtain an adequate and precise quantification of mtLSU rRNA presence in clinical samples, a pre-treatment of the sample should be performed in order to cleavage the DNA and create shorter fragments, improving mtLSU rRNA sequence accessibility and increasing the biosensor sensitivity.”
